# A self-amplifying RNA RSV prefusion-F vaccine elicits potent immunity in pre-exposed and naïve non-human primates

Newly approved subunit and mRNA vaccines for respiratory syncytial virus (RSV) demonstrate effectiveness in preventing severe disease, with protection exceeding 80% primarily through the generation of antibodies. An alternative vaccine platform called self-amplifying RNA (saRNA) holds promise in eliciting humoral and cellular immune responses. We evaluate the immunogenicity of a lipid nanoparticle (LNP)-formulated saRNA vaccine called SMARRT.RSV.preF, encoding a stabilized form of the RSV fusion protein, in female mice and in non-human primates (NHPs) that are either RSV-naïve or previously infected. Intramuscular vaccination with SMARRT.RSV.preF vaccine induces RSV neutralizing antibodies and cellular responses in naïve mice and NHPs. Importantly, a single dose of the vaccine in RSV pre-exposed NHPs elicits a dose-dependent anamnestic humoral immune response comparable to a subunit RSV preF vaccine. Notably, SMARRT.RSV.preF immunization significantly increases polyfunctional RSV.F specific memory CD4$^+$ and CD8$^+$ T-cells compared to RSV.preF protein vaccine. Twenty-four hours post immunization with SMARRT.RSV.preF, there is a dose-dependent increase in the systemic levels of inflammatory and chemotactic cytokines associated with the type I interferon response in NHPs, which is not observed with the protein vaccine. We identify a cluster of analytes including IL-15, TNFα, CCL4, and CXCL10, whose levels are significantly correlated with each other after SMARRT.RSV.preF immunization. These findings suggest saRNA vaccines have the potential to be developed as a prophylactic RSV vaccine based on innate, cellular, and humoral immune profiles they elicit.

Respiratory syncytial virus (RSV), a single-stranded negative-sense RNA virus, is a common cause of severe respiratory disease in children below 5 years and in the elderly, as well as immunocompromised adults (https://www.who.int/teams/health-product-policy-and-standards/standards-and-specifications/vaccine-standardization/respiratory-syncytial-virus-disease). Annually, RSV contributes to significant mortality and morbidity in young children, with one in every 28 deaths during the first 6 months of life in lower- and middle-income countries[1]. Protection of young infants by maternal antibodies from the vaccinated mother wanes over time due to declining antibody titers. Development of effective RSV vaccines has been complicated by the risk of potentially inducing enhanced respiratory disease (ERD) as observed in multiple studies in the 1960's with formalin-inactivated RSV vaccine in infants and RSV seronegative children[2]. This phenomenon was attributed to poor induction of neutralizing antibodies, a Th2-skewed CD4$^+$ T cell response, a dampened type I and II IFN response and eosinophilia in the lung after RSV infection based on animal studies[3]. Murine neonatal RSV

✉e-mail: aneesh.vijayan@gmail.com; rzahn@its.jnj.com

infection models showed that aiding type I and II IFN responses during primary infection resulted in reduced lung pathology during re-infection[4,5]. Additionally, Th1-biased RSV.F vaccines have demonstrated the ability to overcome Th2-bias in animal models, promoting the induction of neutralizing antibodies and T-cells[6,7]. Therefore, vaccination early in life with vaccines capable of eliciting Th1-skewed immune responses to RSV may be important to protect this vulnerable population.

On the other end of the age spectrum, RSV is responsible for an estimated 160,000 hospitalizations and 10,000 deaths in adults aged 65 years and older (https://www.cdc.gov/rsv/high-risk/older-adults.html). Recent market approval of protein-based RSV vaccines for older adults, as well as for pregnant individuals is an important milestone for prevention of RSV-mediated disease[8]. These protein-based vaccines mediate protection primarily via antibodies[9,10]. However, waning efficacy of these vaccines in older adults, especially against lower respiratory tract disease (LRTD), as well as the limited response induced with a booster dose in older adults are concerning. It is hypothesized that reduced RSV-specific functional T-cells correlate with disease severity in older adults[11,12]. Furthermore, a recent study demonstrated the importance of RSV-specific T-cells in controlling infection in the absence of antibodies[13]. Therefore, vaccines that can effectively arm humoral as well as Th1-skewed cellular immunity might be beneficial to elicit durable protection against RSV in elderly individuals.

The ability of RNA-based vaccines, to induce humoral and cellular immune responses, may offer an alternative vaccine platform for the prevention of RSV. Messenger RNA vaccines containing nucleoside-modified bases have gained traction as an effective vaccine platform during the COVID-19 pandemic[14,15]. Likewise, mRNA-1345, a base-modified mRNA vaccine expressing RSV.preF, developed by Moderna®, demonstrate first season efficacy of 83.7% (95.88%CI: 66.1–92.2%; $p < 0.0001$) against RSV-associated LRTD[16]. While T-cell responses induced by the vaccine are not known, efficacy of the vaccine against LRTD drops to 63% in 8.6 months. Next-generation vaccine modalities, based on self-amplifying RNA (saRNA), which can boost potent humoral and cellular immune response in pre-exposed (vaccinated or convalescent) humans may offer an alternative vaccine platform to provide more durable immunity[17,18]. In contrast to base modified mRNA vaccines, the replicative nature of saRNA allows amplification of the gene of interest in vivo. As a result, comparable amount of antigen with relatively longer persistence can be attained at lower dose levels of saRNA in contrast to base modified mRNA[19]. Geall et al. demonstrated that a low dose of LNP formulated saRNA.RSV.preF can significantly reduce lung viral load in vaccinated cotton rats upon RSV challenge[20]. In terms of T-cell induction, research evidence in both animals and humans demonstrated that saRNA can stimulate the generation of a de-novo response or enhance pre-existing responses against specific antigens[17,21]. Using low dose levels of vaccine has several benefits from increasing the number of vaccine doses to mixing different antigens for a multivalent vaccine. In a recent phase 3 study evaluating COVID-19 booster vaccination efficacy, a low-dose (5 mcg) saRNA COVID-19 vaccine elicited an effective response surpassing the response of a standard booster dose (30 mcg) of mRNA-COVID-19 vaccine, thereby paving the way to approval in Japan[22,23]. Additionally, cellular responses induced by a saRNA-based COVID-19 vaccine tend to be skewed towards Th1 and remain durable for at least 6 months in individuals who have been previously exposed to the antigen[17,22].

To investigate the immunogenicity of a saRNA RSV vaccine, a Venezuelan equine encephalitis virus (VEEV)- based saRNA encoding the prefusion stabilized F (pre-F) protein of RSV, similar to the clinical tested antigen encoded by Ad26.RSV.preF[24], was constructed and formulated into lipid nanoparticles (SMARRT.RSV.preF). Here, we report pre-clinical evaluation of the vaccine in naïve NHPs and mice as well as in RSV pre-exposed NHPs to mimic the exposure status in older

adults, the anticipated target population. Our data supports the clinical potential of a single dose of LNP-formulated saRNA RSV vaccine to boost humoral and cellular immune responses in a pre-exposed setting. In addition, we demonstrate that SMARRT.RSV.preF vaccine induces systemic cytokines/chemokines in NHPs that are associated with a type I IFN response.

## Results

### Design and in-vitro characterization of SMARRT.RSV.preF vaccine

saRNA expressing full-length, membrane-anchored, prefusion stabilized protein of RSV A2 strain was formulated in LNP with ALC-0315 cationic ionizable lipid (SMARRT.RSV.preF). ALC-0315 is one of the key components of Comirnaty®, a base modified RNA-based COVID-19 vaccine, and has been tested extensively in humans[15]. To circumvent the impact of host innate response on replicon translation, the SMARRT platform includes an RNA motif called downstream loop (DLP) from Sindbis virus placed upstream of non-structural protein 1 (nSP-1) (Fig. 1a)[25]. SMARRT.RSV.preF had an average hydrodynamic particle size of $75 \pm 5$ nm with polydispersity index (PDI) of <0.2. Biophysical measures remained unaffected after one cycle of freeze-thaw suggesting particle stability (Supplementary Table 1). Total and surface expressed membrane-anchored RSV.F was confirmed with an anti-RSV.F specific monoclonal antibody in BHK cells transfected with the vaccine (Fig. 1b, c).

### SMARRT.RSV.preF elicits neutralizing antibodies and polyfunctional T-cells in mice

Immunogenicity of SMARRT.RSV.preF vaccine was evaluated in BALB/c mice ($n = 8$/group), in a two-dose intra-muscular regimen (28 days apart) with dose levels of 0.1 mcg, 1 mcg and 10 mcg. As a positive control arm, we included a group that received a single dose of $10^{10}$ viral particles (VPs) of Ad26.RSV.preF (Fig. 2a). A dose-dependent increase in RSV.CL57 neutralizing antibodies was detected in immunized animals after 1-dose with GMT of 562 for 0.1 mcg, 1060 for 1 mcg and 10,369 for 10 mcg dose levels of SMARRT.RSV.preF (Fig. 2b). After a 2nd dose at day 28, RSV.CL57 neutralizing antibodies (VNA) increased by 20- to 5-fold with increasing dose levels of SMARRT.RSV.preF. Kinetics and dose-dependent induction of RSV.preF binding antibodies followed the same trend as neutralizing antibodies (Fig. 2c). As expected, all animals seroconverted after a single dose of Ad26.RSV.preF. To assess cellular response, splenocytes from animals immunized with 2-doses of 10 mcg SMARRT.RSV.preF were stimulated with RSV.F peptide pool to detect IFNγ, TNFα and IL2 positive CD8+ and CD4+ T-cells by intra-cellular cytokine staining assay (ICS). SMARRT.RSV.preF elicited IFNγ+ and TNFα+ monofunctional as well as IFNγ+TNFα+IL2+ and IFNγ+TNFα+ polyfunctional RSV.F specific CD8+. In addition, low frequencies of monofunctional CD4+ T-cells as well as IFNγ+TNFα+IL2+ and IFNγ+TNFα+ and TNFα+IL2+ double positive cells were elicited by SMARRT.RSV.preF (Fig. 2d). Taken as a whole, these results demonstrated that SMARRT.RSV.preF is immunogenic in mice after one immunization and a 2nd dose can further increase the response.

### SMARRT.RSV.preF can boost neutralizing antibodies in RSV pre-exposed NHPs and elicits a de-novo humoral response in RSV naïve NHPs

The potency of the SMARRT.RSV.preF vaccine to induce humoral and cellular immunity was assessed in naïve as well as RSV pre-immune setting (Fig. 3a). To mimic pre-immune condition, a cohort of 12 NHPs was infected with $10^6$ PFU of RSV.A2 or RSV$_{Memphis37}$ via the intranasal and intratracheal route. Three months later, NHPs were vaccinated with 1 mcg or 10 mcg SMARRT.RSV.preF or with 50 mcg of recombinant RSV.preF protein (PRPM) at week 0. A comparator arm of RSV naïve ($n = 4$) animals received 10 mcg of SMARRT.RSV.preF vaccine at week 0. Humoral immune responses were quantified by determining

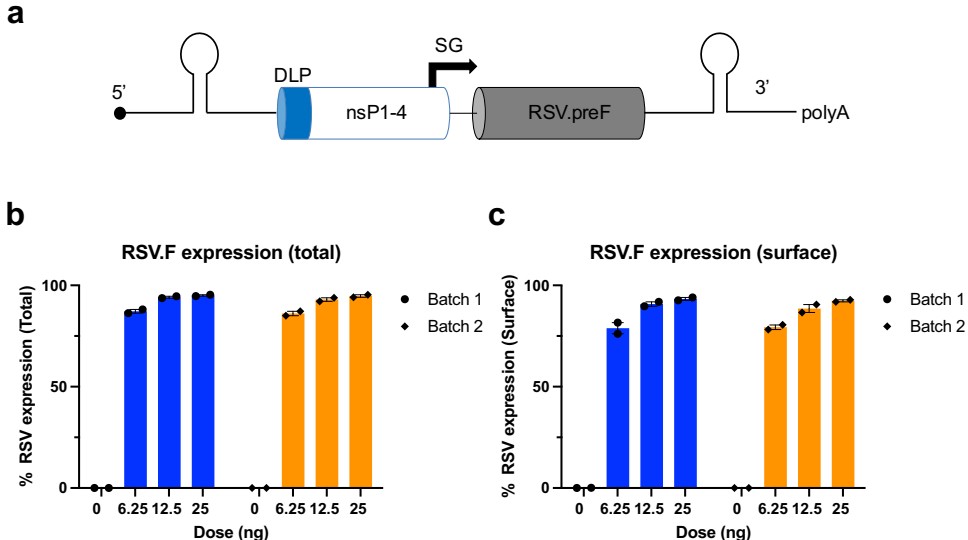

**Fig. 1 | SMARRT.RSV.preF design and in vitro activity. a** Design of SMARRT.RSV.preF. The conserved sequence element (CSE), needed for replication, downstream loop (DLP), to promote translation, non-structural proteins 1–4, which form the replication complex, sub genomic promoter (SG) to drive sub genomic RNA synthesis, transgene region encoding prefusion stabilized RSV F, and poly A tail (schematics not to scale). **b** Total and **c** Surface expression of RSV.F protein in BHK cells 24 h post-transfection with LNP formulated SMARRT.RSV.preF. Data for two separate batches of formulated material are presented as mean percentage of cells expressing RSV.preF protein ± SEM of $n = 2$ (closed circles) biological replicates tested for each concentration of LNP formulated SMARRT.RSV.preF. Source data are provided as a Source Data file.

neutralizing antibodies against RSV.CL57 (IC50) as well as RSV.preF specific IgG binding antibodies.

Anamnestic VNA responses in RSV infected NHPs, at week 2 compared with week 0, was significantly higher after a single dose of PRPM (GMT VNA$_{IC50}$: 14664; $p < 0.0001$), 1 mcg (GMT VNA$_{IC50}$: 2336; $p = 0.0003$) and 10 mcg of SMARRT.RSV.preF (GMT VNA$_{IC50}$: 5873; $p < 0.0001$) (Fig. 3b). The peak geometric mean fold rise (GMFR) of the anamnestic VNA response was 143-, 33- and 99-fold in PRPM, 1 mcg or 10 mcg SMARRT.RSV.preF vaccine groups respectively compared with baseline titer. In naïve animals, a single dose of 10 mcg of SMARRT.RSV.preF raised VNA titers by week 2 (GMT VNA$_{IC50}$: 152; $p = 0.1155$). While peak anamnestic VNA titers were not statistically significantly different between 1 mcg and 10 mcg doses of SMARRT.RSV.preF vaccine in the RSV pre-exposed animals ($p = 0.0920$), titers induced by 10 mcg dose of vaccine were 16-fold higher in the RSV pre-exposed animals than in naïve animals dosed with the same dose of the vaccine ($p = 0.0004$). During the 3-month follow-up period, we observed a gradual decrease in the VNA response for PRPM (reduced by 7-fold), 1 mcg SMARRT vaccine (reduced by 4-fold), and 10 mcg SMARRT.RSV.preF (reduced by 6-fold). However, these responses remained significantly higher than the level prior to vaccination. This is also evident in the overall responses, estimated by analysis of area under the curve (AUC) (Fig. 3c). The levels of RSV A2 plaque reducing neutralizing antibody titers (PRNT) showed a similar trend as VNA (Supplementary Fig. 1). An identical trend was observed for RSV.preF specific serum IgG binding antibodies with a peak response at 2-week post-immunization with SMARRT.RSV.preF in RSV infected animals (Fig. 3d). However, the maximal levels of preF antibodies in RSV infected animals induced by 10 mcg of SMARRT.RSV.preF were 5-fold higher than by the 1 mcg dose ($p = 0.0360$).

To further evaluate the effectiveness of SMARRT.RSV.preF in a booster context, animals that received the vaccine as a first dose were injected with 1 mcg of SMARRT.RSV.preF at week 16. The vaccine boosted RSV A2 PRNT levels reached the upper limit of quantification in the majority of RSV-infected animals independent of the dose level used for the first dose. In naïve animals that received 10 mcg of SMARRT.RSV.preF as the initial dose, PRNT levels rose by 22-fold 2 weeks post boosting (Supplementary Fig. 1).

Studies have shown the association of RSV-specific IgA mucosal antibodies to protection in humans and NHPs[26,27]. Therefore, we measured RSV.preF specific IgA present in nasal swabs collected at week 8 post-primary immunization. All RSV-pre-exposed animals dosed with 10 mcg of SMARRT.RSV.preF and 3 out of 4 animals dosed with 1 mcg SMARRT.RSV.preF had detectable IgA antibodies in comparison with naïve animals immunized with SMARRT vaccine (1 out of four) (Fig. 3e). The mucosal RSV.preF IgA responses between naïve and RSV infected NHPs following SMARRT vaccination were statistically significantly different ($p = 0.0275$). Only 2 out of 4 animals had detectable RSV.preF IgA antibodies post-PRPM immunization. Overall, these data demonstrate that SMARRT.RSV.preF vaccination induced an anamnestic serological response in RSV infected NHPs accompanied by detectable levels of RSV.preF IgA antibodies in the nasal compartment.

## Poly-functional RSV.F specific CD4$^+$ and CD8$^+$ T-cells are induced by SMARRT.RSV.preF vaccine

Having demonstrated the potential of SMARRT.RSV.preF to elicit humoral responses, we studied antigen-specific cellular responses in peripheral blood mononuclear cells (PBMCs) isolated from immunized non-human primates via an IFNγ ELISpot assay. PRPM protein did not increase RSV.F specific IFNγ ELISpot responses in RSV pre-exposed animals after immunization (Fig. 4a). Immunization with 1 mcg and with 10 mcg of SMARRT.RSV.preF led to a GMFR of 6.5-fold ($p = 0.0291$) and 8.2-fold ($p = 0.004$) at week 4 compared to baseline, respectively. Furthermore, in RSV-infected animals, the peak ELISpot response was higher and more consistent in all animals dosed with 10 mcg of SMARRT.RSV.preF (GMR = 3.07 ± 0.1Log$_{10}$ SFU/10$^6$ PBMCs) compared to the 1 mcg dose (3 out of 4 animals responded; GMR = 2.59 ± 0.36Log$_{10}$ SFU/10$^6$ PBMCs). In contrast, a de-novo response of 2.4 ± 0.19Log$_{10}$ SFU/10$^6$ PBMCs was elicited by 10 mcg of SMARRT.RSV.preF vaccine in naïve animals at week 4. Three months post immunization, the responder rate in naïve NHP was 50% after

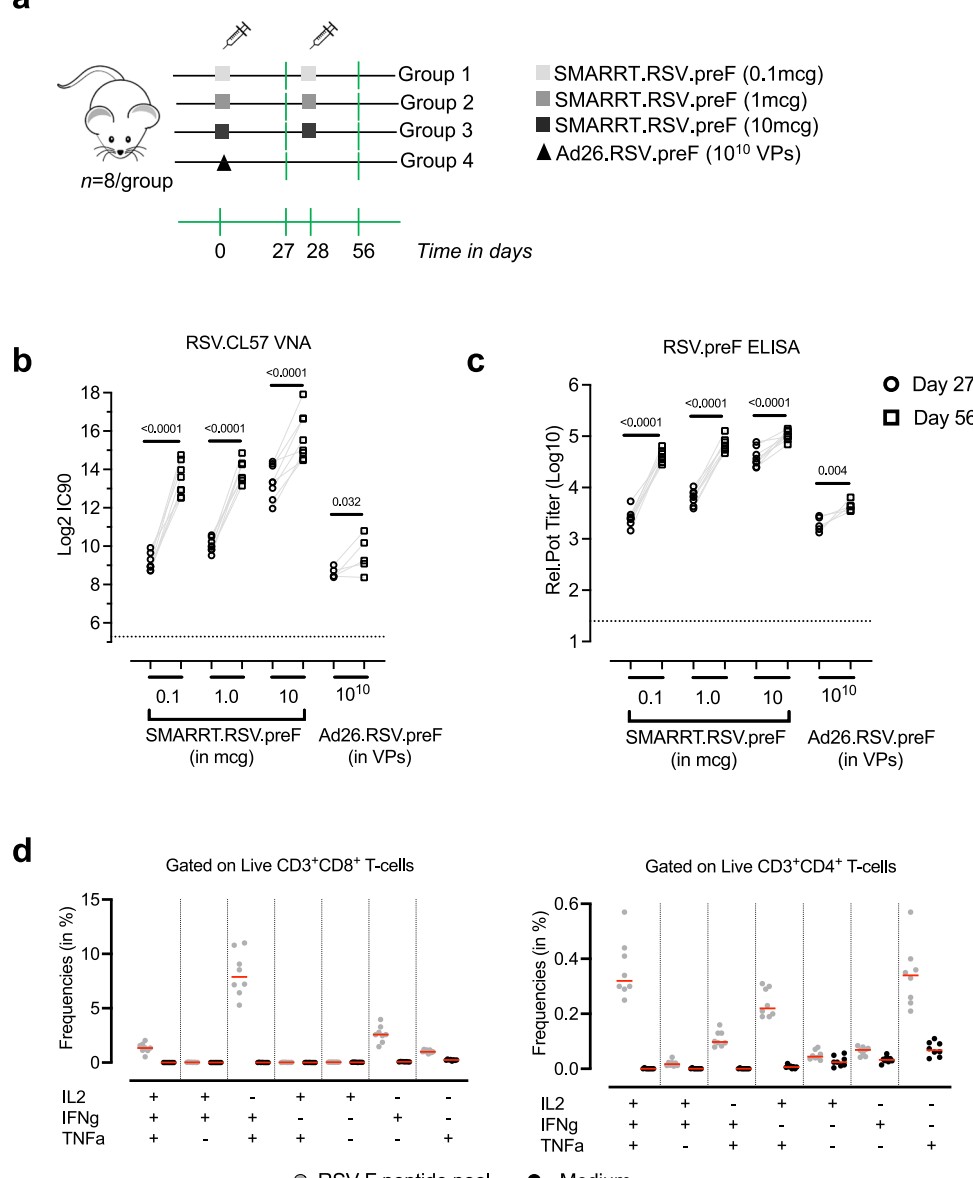

**Fig. 2 | SMARRT.RSV.preF vaccination elicits humoral and cellular response in mice. a** Vaccines SMARRT.RSV.preF (0.1, 1, and 10 mcg) or Ad26.RSV.preF (10[10] VPs) were administered intramuscularly to female BALB/c mice. While Ad26.RSV.-preF was given as a single dose, SMARRT vaccine was administered as a 2-dose homologous regimen given 4 weeks apart. Components of this figure were sourced from Open Clip Art, under a Creative Commons Attribution 3.0 Unported License; https://creativecommons.org/licenses/by/3.0/. The serum from *n* = 8 biologically independent animals per group was sampled on days 0, 27 and 56. **b** RSV.CL57 neutralizing antibody titers and **c** RSV.preF specific IgG antibodies in the serum of immunized animals measured at day 27 (open circles) and day 56 (open squares).

Limit of detection (LoD) is represented by the dotted line. Gray lines represents paired measurements. Statistical comparisons of day 27 to day 56 humoral responses were determined with ANOVA and adjusted for multiple comparisons with Bonferroni correction of log transformed data. **d** RSV.F specific T-cells measured in the spleens of 10 mcg SMARRT.RSV.preF immunized animals (*n* = 8 biologically independent animals), at day 56, by ICS. Different subsets of IFNγ, TNFα and IL2 positive CD8+ and CD4+ T-cells gated on Live CD45+ cells. Gray circles represent splenocytes stimulated with RSV.F peptide pool and black circles represent corresponding medium stimulated samples. Red horizontal lines represent median response. Source data are provided as a Source Data file.

dosing of induced by a 10 mcg dose of SMARRT.RSV.preF vaccine in 50% of naïve NHPs. By contrast, the response in RSV-infected animals immunized with an equivalent dose of the vaccine gradually declined in all animals but was still 3-fold higher than the pre-immunization response levels three months post-vaccination (*p* = 0.0287) in all animals.

Systemic RSV.F specific polyfunctional T-cells are positively correlated with lower rates of re-infection in humans[28]. To assess magnitude and polyfunctionality of RSV.F-specific memory T-cells in PBMCs, an ICS assay was conducted at baseline and week 4. Consistent with the

ELISpot response, PRPM immunization did not result in an increase in RSV.F specific memory CD4+ and CD8+ T-cells in RSV pre-exposed NHP. However, following SMARRT.RSV.preF immunization, RSV infected non-human primates (NHPs) exhibited elevated levels of RSV.F specific memory CD4+ and CD8+ T-cells at week 4 compared to baseline (Fig. 4b). Although higher-dose SMARRT.RSV.preF immunization led to a higher number of T-cells than the low-dose group in RSV infected NHPs, the difference did not reach statistical significance. Among naïve animals, a slight increase in memory CD4+ T-cells was observed after 10 mcg SMARRT.RSV.preF vaccination, with minimal impact on CD8+

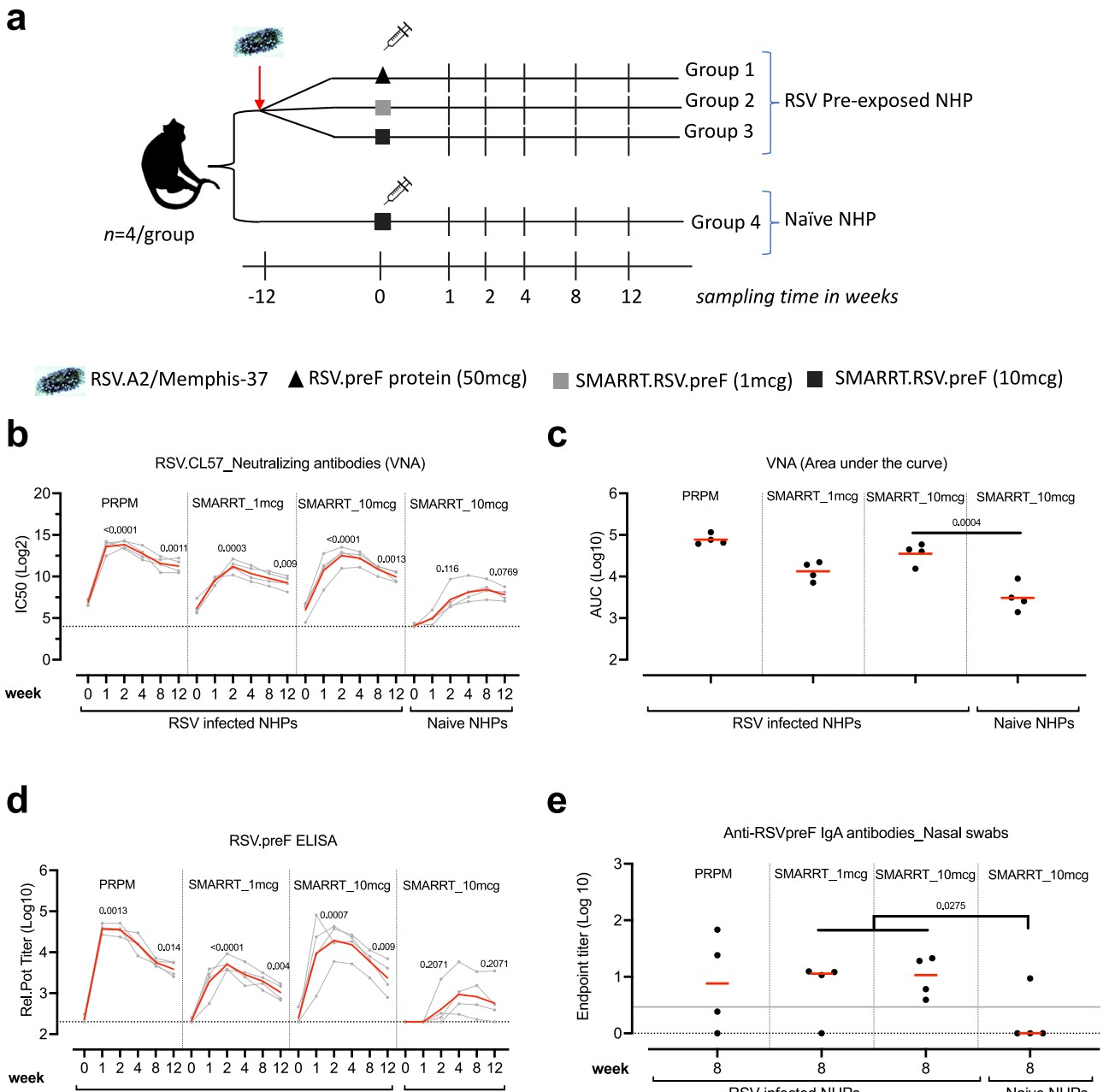

**Fig. 3 | Humoral immunogenicity elicited by SMARRT.RSV.preF vaccine in RSV infected and in naïve NHPs. a** Cynomolgus NHP (*n* = 12) infected with RSV at week −12 were distributed into 3 groups based on RSV.postF antibody titers measured at week −8. At week 0, pre-exposed animals (*n* = 4/group) received an intra-muscular immunization with either 50 mcg of PRPM (group 1) or 1 mcg (group 2); 10 mcg (group 3) of SMARRT.RSV.preF. A RSV naïve group (*n* = 4) received 10 mcg of SMARRT.RSV.preF vaccine (group 4). PBMCs and serum samples were collected from *n* = 4 biologically independent animals per group at weeks 1, 2, 4, 8, and 12 in addition to nasal swabs (week 8). Components of this figure were sourced from Google Images, under a Creative Commons Attribution 2.0 Unported License; https://creativecommons.org/licenses/by/2.0/. **b** RSV.CL57 neutralizing antibodies; **c** AUC for RSV.CL57 VNA and **d** RSV.preF IgG antibodies determined for each animal (gray or black circles) from week 0 through week 12 of the study. The geometric mean titers (GMT) and LoD for each group is represented by the red horizontal line and the dotted line. **e** RSV.preF specific IgA antibodies measured in nasal swab elutes (*n* = 4/group). Samples were only included if they were free of blood contamination, with maximum number of uncontaminated samples available from weeks 0 and 8. Measurements were corrected for total protein content in elutes. The dotted line represents the LoD while the gray solid line represents the upper limit of 95% CI of mean response in RSV-infected animals measured at week 0 prior immunization. Statistical significance was determined with ANOVA and adjusted for multiple comparisons with Bonferroni correction between (**c**, **e**) RSV-infected and naïve animals following SMARRT.RSV.preF immunization (**b**, **d**) between responses at weeks 2 and 12 to week 0. Source data are provided as a Source Data file.

T-cells. Notably, a three-fold increase in memory CD4$^+$ T-cells was detected in the RSV infected group compared to the naive group following administration of 10 mcg of SMARRT.RSV.preF vaccine (*p* = 0.0320). While a similar trend was also observed for CD8$^+$ T-cells, it did not reach statistical significance (2.2-fold; *p* = 0.3694).

To assess differences in the quality of T-cell responses, polyfunctionality scores (PFS) were summarized using the COMPASS tool[29]. An overall trend of increased PFS at week 4 compared to baseline was observed, regardless of the vaccine administered. For a deeper analysis of T-cell responses, SPICE was employed to examine

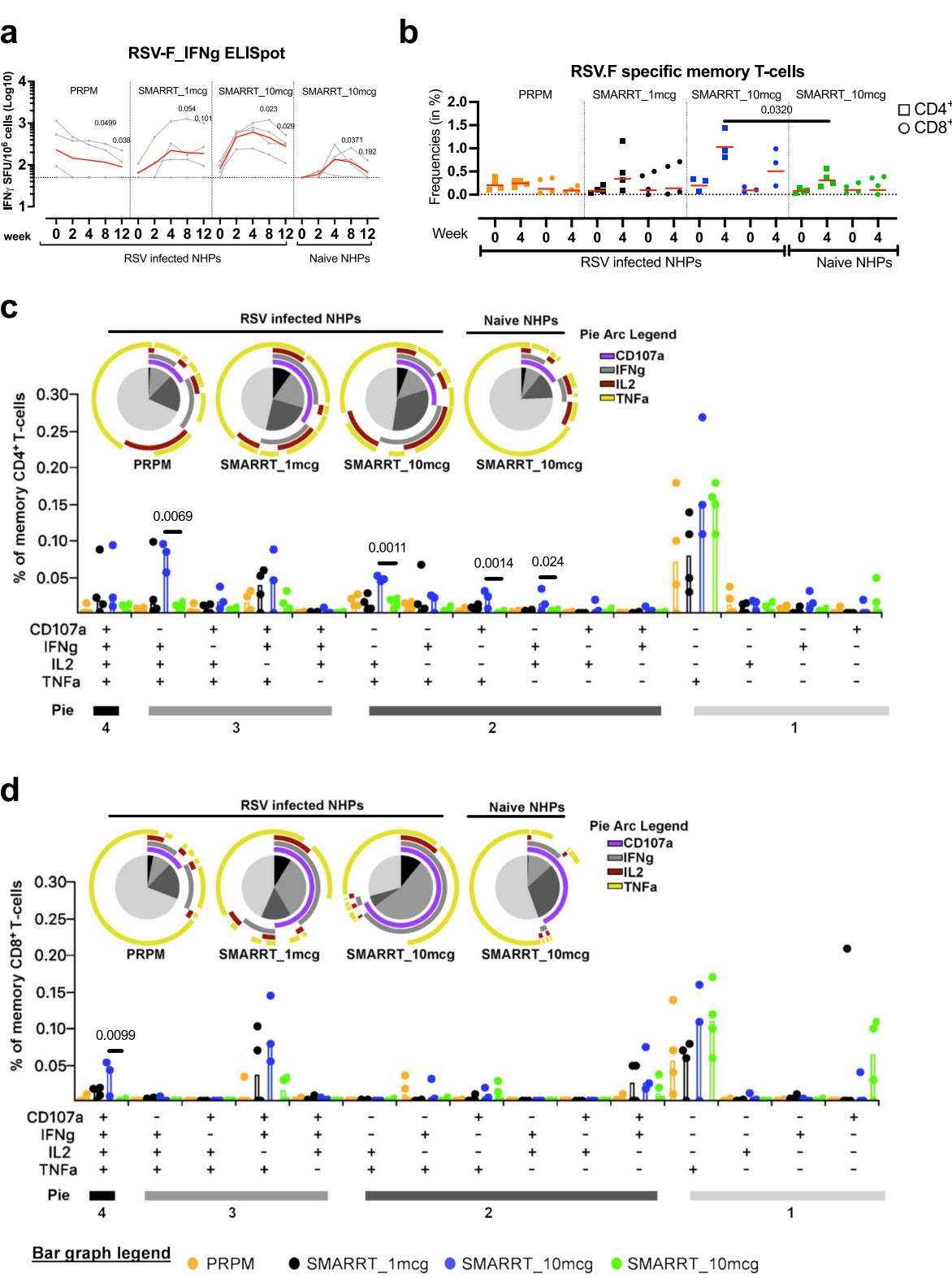

the T-cell polyfunctionality induced by the vaccines at week 4[30]. Post-immunization, the protein vaccine did not noticeably influence the magnitude of RSV.F specific T-cells (Fig. 4b and Supplementary Fig. 2). With the PRPM vaccine, only one-fourth of the memory CD4+ and CD8+ T-cells exhibited polyfunctionality, with a dominant mono-functional population positive for TNFα (Fig. 4b, c). In contrast, RSV-infected animals receiving either the 1 mcg or 10 mcg dose of

SMARRT.RSV.preF displayed greater than 50% polyfunctional CD4+ and CD8+ T-cells (Fig. 4c, d; pie charts). While a tendency for a higher magnitude of polyfunctional T-cells was noted with the 10 mcg dose compared to the 1 mcg dose of SMARRT.RSV.preF, the differences in these T-cell subsets did not reach statistical significance (Fig. 4c, d; bar charts). Polyfunctional responses in naive NHPs administered with 10 mcg dose of SMARRT.RSV.preF were significantly lower compared

**Fig. 4 | Magnitude and polyfunctionality of cellular responses elicited by SMARRT.RSV.preF in NHPs. a** RSV.F specific IFNγ secreting cells in PBMCs of immunized animals (*n* = 4 biologically independent animals per group) represented as number of Spot Forming Units (SFU) per million cells. The dotted line represents the positivity threshold of the assay which is set at 50 SFU and the red line indicates the geometric mean response. Background subtracted response of each animal (gray circles) overtime is shown. Comparisons were made between baseline response to weeks 8 and 12. **b** CD4 (square) and CD8 (circle) memory T-cells gated on Live CD45⁺CD28±D95⁺ positive for CD107a or IFNγ or TNFα or IL2 stimulated with RSV.F peptide pool were identified using intracellular cytokine staining. After subtracting the corresponding response in the medium stimulated sample of each animal, values are shown with a threshold represented by the dotted line and median response by the red line. Due to insufficient number of cells, data from an

RSV pre-exposed animal immunized with 10 mcg SMARRT.RSV.preF group is not available (**b**–**d**). Polyfunctional subsets of CD4 and CD8 T-cells by Boolean gating and after background subtraction were subsequently analyzed using SPICE at week 4 post-immunization. Bar plots represent background-subtracted median frequency of cells in each subset for each animal (circles). Pie chart wedges represent the functional subsets producing different combinations of cytokines i.e., 4+, 3+, 2+ and monofunctional subsets indicated by the color coding under "Pie", while the surrounding pie arcs represent total median level of each analyte. Statistical analysis was done on square root transformed values. **b**–**d** Comparisons between SMARRT.RSV.preF immunized groups was determined with ANOVA (TOBIT model) and adjusted for multiple comparisons with Bonferroni correction. Source data are provided as a Source Data file.

to the response in RSV infected animals. In fact, frequency of memory CD4⁺ IFNγ⁺TNFα⁺IL2⁺ (0.1%) triple positive population in RSV infected animals was ten times higher than in naïve NHPs after SMARRT.RSV.preF administration (Fig. 4c). Notably, the proportion of memory CD4⁺ cells contributing to IFNγ and CD107a were higher after vaccination with either 1 mcg or 10 mcg SMARRT.RSV.preF in RSV-infected animals compared to naïve animals (pie arcs). Within the memory CD8⁺ T-cell compartment, CD107a⁺IFNγ⁺TNFα⁺IL2⁺, CD107a⁺IFNγ⁺TNFα⁺ and CD107a⁺IFNγ⁺ cells accounted for 50% of RSV.F specific CD8⁺ T-cells in RSV primed animals immunized with SMARRT.RSV.preF, irrespective of the dose levels (Fig. 4d). These polyfunctional cells contributed the most to CD107a and IFNγ levels (pie arcs). In contrast, SMARRT.RSV.preF immunization in naïve animals induced mainly monofunctional memory CD8⁺ T-cells for CD107a and TNFα. Thus, a single dose of SMARRT.RSV.preF vaccine induce durable cellular responses of higher magnitude and improved quality over a protein-based RSV vaccination in previously RSV-infected animals.

### SMARRT.RSV.preF vaccine induces inflammatory cytokines involved in immune cell chemotaxis

The early innate immune response after vaccination is known to shape adaptive responses[31,32]. Here, we measured 45 analytes (inflammation-related cytokines/chemokine) in the serum pre- and 24 h post-vaccination. Principal component analysis (PCA) was used to determine factors that explain the maximum variance in the dataset (Fig. 5a). The most variance in analyte levels is explained by the sampling timepoint (0 h vs 24 h) and dose level (1 mcg vs 10 mcg) of SMARRT.RSV.preF. Based on linear modeling with PCA, no clear distinction could be made between animals dosed with 10 mcg SMARRT.RSV.preF vaccine in RSV infected or naïve animals. Therefore, these groups were pooled together for downstream analysis.

While PRPM immunization did not result in any change in the levels any of the analytes, nine and twenty-three analytes showed statistically significant changes at 24 h after post-vaccination with 1 mcg and 10 mcg SMARRT.RSV.preF compared with baseline levels. Differential expression visualization of these analytes post dosing with 1 mcg and 10 mcg of SMARRT.RSV.preF is shown in Fig. 5b. Elevated levels of pro-inflammatory analytes such as IL6, TNFα, IL18, IL15, VEGF-A and analytes involved in immune cell trafficking/modulation such as CXCL9, CXCL10, CXCL11, CSF1, Flt3LG, CCL-3, 4, 8, 11, and 19 were observed. Interestingly, downregulated analytes included EGF and TGFα which are ligands for Epidermal Growth Factor Receptor (EGFR) and MMP-1 and MMP-12 which are known to play important roles in cell migration and angiogenesis[33]. A similar pattern was observed for 1 mcg of SMARRT.RSV.preF vaccine, however a lower inflammatory response reduced the number of analytes above the threshold.

To understand the relationship between analytes, we performed an inter-analyte correlation analysis. A distinct cluster of cytokine/chemokines comprising IL15, CCL4, CXCL10 and TNFα had high inter-correlation suggesting their coordinated role in modulating innate responses induced by SMARRT.RSV.preF (Fig. 5c). Correlation matrix

also revealed a positive association of Flt3LG and CCL3 aside from the above-mentioned cluster. Additional positive hits such as IL6-TNFα ($r = 0.74$) associated with inflammation, CSF1-CCL8-CCL2 ($r = 0.92$) axis involved in macrophage chemotaxis and differentiation and CXCL10-CXCL11 ($r = 0.83$) ligands for CXCR3 expressing immune cells were observed.

To investigate biological processes or pathways influenced by cytokine/chemokines an enrichment analysis on analytes was performed. The analysis identified canonical pathways associated with inflammation mediated by interferon (type I and II) signaling pathways (Supplementary Table 2). Altogether, SMARRT.RSV.preF vaccination induced a dose-dependent increase in pleiotropic cytokines/chemokines mainly involved in inflammation and immune cell chemotaxis.

## Discussion

In this study, we present the results of our nonclinical evaluation of a LNP formulated saRNA vaccine expressing RSV.preF protein. We demonstrate that a single dose of SMARRT.RSV.preF elicited potent humoral and cellular responses in mice and in naïve and RSV pre-infected NHPs. To our knowledge this study is the first to extensively characterize the immunogenicity of a saRNA-based RSV vaccine concurrently in both naive and RSV-infected NHPs. Our findings show that in an RSV pre-exposed setting, the humoral response elicited by the SMARRT.RSV.preF vaccine is comparable to that of a pre-fusion stabilized RSV.F protein vaccine, which is currently the standard of care. In contrast to the protein vaccine, SMARRT.RSV.preF immunization enhanced the magnitude and quality of cellular responses, but also stimulated an innate response that is biased towards Th1 in NHPs. Overall, these results highlight inherent differences in the immunogenicity of LNP-formulated saRNA-based and protein-based RSV vaccines in RSV-infected NHPs, with the main differentiation being the induction of polyfunctional cellular responses being a noteworthy feature by LNP-formulated saRNA immunization.

It has been shown that the waning efficacy of current RSV vaccines, especially against mild to moderate LRTD, corresponds with declining levels of serum antibodies from peak[34]. Both SMARRT.RSV.preF and PRPM vaccines increased RSV VNA by more than 90-fold at peak response and were sustained through week 16 above baseline. Similar fold increase in RSV VNA has been reported previously with protein and other RSV.preF expressing vaccine modalities in NHPs[35,36]. By contrast, in adult humans, some of these RSV vaccines increased RSV neutralizing antibodies by a factor of 5 to 20-fold at peak[10,16]. While the observed discrepancy will make it difficult to directly translate our findings in NHPs to humans, including the durability of the humoral response, the rapid rise in VNA as early as one week following immunization with SMARRT.RSV.preF in RSV infected versus naïve animals suggest an ability of this investigational vaccine to raise an anamnestic humoral response.

The post boost response induced by 1 mcg of SMARRT.RSV.preF, is notable. This dose is relatively low compared to the standard booster dose of prophylactic mRNA vaccines in humans (ranging from 30 mcg

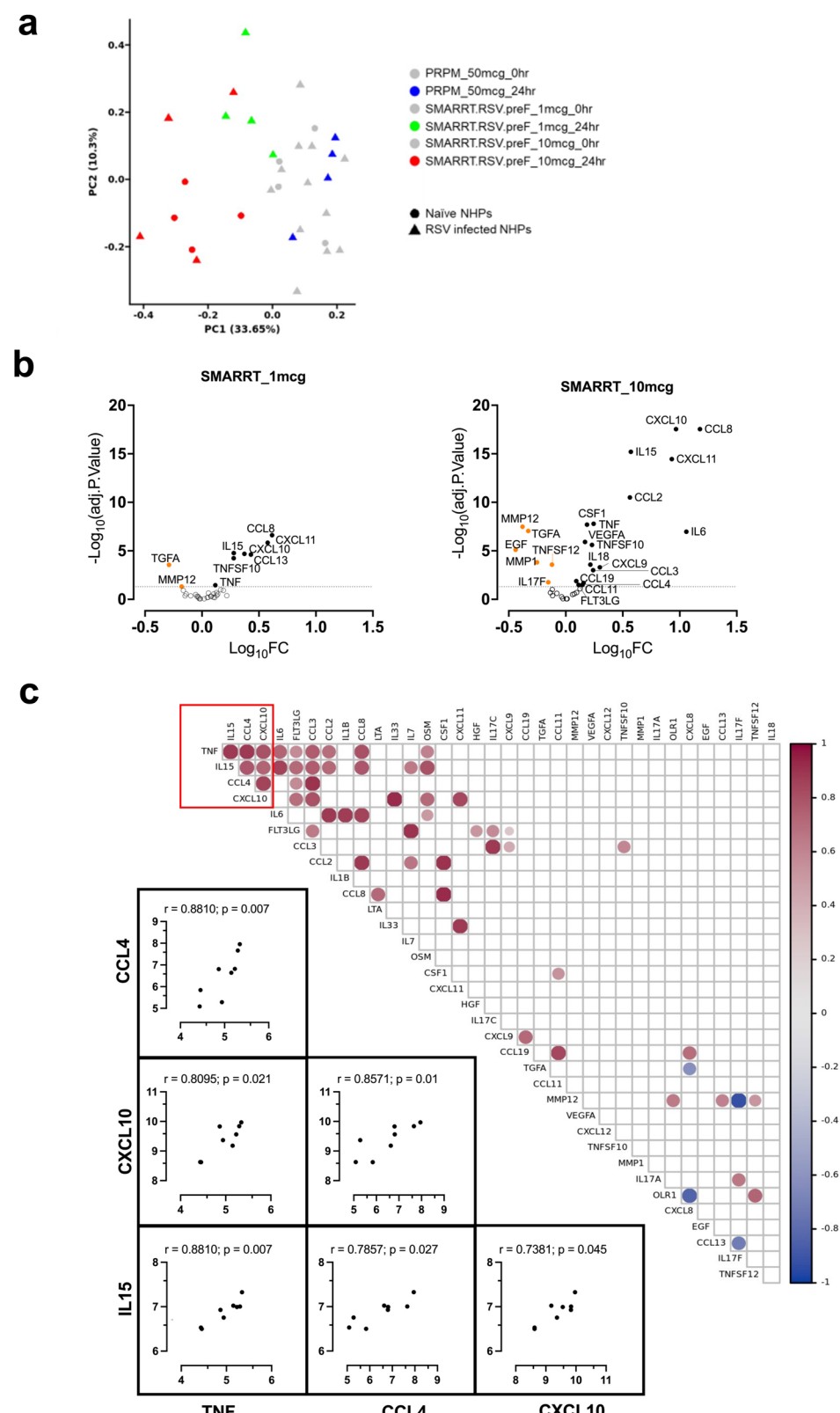

to 100 mcg). This, highlights the dose-sparing capability of saRNA vaccines. Preclinical studies and human trials indicate that a typical dose range for a prophylactic saRNA-based vaccine is between 1–10 mcg, which is consistent with the doses tested in this study[18,21]. Another aspect which we did not explore in this study is the impact of intervals between SMARRT.RSV.preF dosing on immune responses. Clinical data suggests that a longer interval (>14 weeks) between

saRNA doses resulted in higher humoral response compared with shorter intervals (4 weeks)[18]. This presents opportunities to not only utilize the SMARRT.RSV.preF vaccine for seasonal boosters but to also reduce booster dosage requirements increasing vaccine availability and cost-effectiveness. A limitation of the current study is the durability of humoral responses elicited by SMARRT.RSV.preF which was not continuously monitored. Given the recent clinical evidence

**Fig. 5 | Serum chemokine/cytokine profiling from NHPs immunized with SMARRT.RSV.preF.** Analytes were measured in the serum of immunized animals ($n = 4$ biologically independent samples per group) prior to immunization and at 24 h post immunization. **a** Principal component analysis of analytes by treatment and sampling timepoints. Color represents the vaccines and corresponding timepoints while shapes represent the pre-exposure status of animals to RSV. **b** Volcano plots showing significance (-log10 of adjusted p-value) versus log10 fold changes of analytes at 24 h post-immunization with 1 mcg and 10 mcg of SMARRT.RSV.preF vaccine compared to baseline. Statistical comparisons of analyte concentrations between treatment groups were conducted using a mixed effects model (moderated two-sided t-statistics) by the limma package. The gray dotted line shows a significant threshold of adjusted p-value = 0.05 with analytes that are significantly upregulated indicated by the closed black circles and those that are significantly downregulated represented by orange closed circles. Analytes that fall below the significance threshold are indicated by open circles below the dotted line. **c** Correlations of log10 analyte concentrations presented as a heatmap correlogram in animals dosed with 10 mcg of SMARRT.RSV.preF. Spearman rank correlation coefficients ($r$) between each pair of analytes with significant ($p < 0.05$) non-adjusted $p$-values are shown in the correlation matrix with the color intensity of the circles indicating the direction and degree of correlation. The red box identifies a cluster of analytes namely CXCL10, IL15, TNF and CCL4 that have a pairwise positive correlations with each other. Individual, two-tailed, Spearman correlation plots of these analytes in this cluster for each animal are shown with Spearman coefficient values ($r$) and non-adjusted p-values. Source data are provided as a Source Data file.

demonstrating sustained humoral responses generated by saRNA vaccines compared to mRNA vaccines, further studies to evaluate the durability of humoral response elicited by SMARRT.RSV.preF would be informative[22].

Studies suggest that mucosal humoral factors, particularly the presence of nasal RSV.F IgA antibodies, are associated with protection against RSV upper and lower respiratory tract infection[26,27]. In this study, a single dose of SMARRT.RSV.preF, independent of dose levels, increased RSV.preF IgA antibodies in the nasal compartment in RSV-primed animals but failed to induce a de novo response in a majority of naïve animals. This is in line with studies that demonstrated the ability of mRNA-based COVID-19 vaccines to induce mucosal IgA response in convalescent individuals but not in naïve subjects after one dose[37,38]. The differential ability of mRNA/saRNA vaccines to induce mucosal IgA responses in convalescent versus naïve subjects maybe due to boosting pre-existing mucosal resident B-cells generated during viral infection or primary vaccination via circulating antigen or migration of transduced cells from site of immunization[39].

From human challenge studies, there is increasing evidence suggesting T-cells play a critical role in mediating protection against RSV[12,13]. The development of RSV antigen-specific T-cells through vaccination could have potential benefits in reducing the severity of the disease, especially in cases where antibodies may be ineffective due to antigenic variability among different circulating strains[11,40]. Revaccination with protein vaccines, after a year, was found to increase CD4+ T-cells to levels like those observed after the first dose in older adults[41]. However, the polyfunctionality of these cells has not been extensively studied, although they appear to be primarily Th1 biased. In our study, we observed that RSV infection in NHPs significantly increased polyfunctionality scores of memory CD4+ T-cells suggesting induction of polyfunctional cells (Supplementary Fig. 2). The observation is consistent with the data from a controlled RSV human challenge study where a limited number of polyfunctional RSV-specific CD4+ T-cells (IFNγ+TNFα+) were detected with very low CD8+ T-cells[42]. Our data shows the potential of SMARRT.RSV.preF to induce polyfunctional memory CD4+ and CD8+ T-cells. Notably, SMARRT.RSV.preF generated CD8+ T-cells, expressing CD107a, a degranulation marker for cytolytic T-cells, and IFNγ that are known to be lower in older individuals susceptible to RSV infection[11]. By contrast, protein- or mRNA-based RSV-F vaccines do not elicit CD8+ T-cells at any point post-immunization in pre-clinical models or in humans[34,35,43]. Additionally, CD4+ T-cells elicited by SMARRT.RSV.preF vaccine produced IFNγ and TNFα cytokines consistent with a Th1 response, which offers potential advantage for active immunization in infants and children, where a Th1 skewed RSV vaccine is needed[44]. We speculate that these polyfunctional T-cells induced by vaccination could be recalled rapidly to accelerate virus clearance thereby counteracting lower antibody titers[13,28]. While we acknowledge that the use of a non-adjuvanted protein and variations in the presentation of RSV.preF protein (soluble vs membrane bound by the RNA vaccine) prevents direct comparison of vaccine platforms, the ability of SMARRT.RSV.preF to enhance both

memory CD4+ and CD8+ polyfunctional T-cell responses compared to the protein vaccine is promising. It should be noted that our study did not evaluate mucosal tissue resident RSV.F specific T-cells ($T_{RM}$) generated by SMARRT.RSV.preF vaccine. However, previous literature suggests that the saRNA platform, which our investigational vaccine utilizes, has the potential to induce $T_{RM}$ in preclinical models, which could be a significant advancement in the field of RSV vaccines[45].

Investigation into the strong innate response caused by the replication activity of the saRNA backbone versus LNP formulation and each component impact on antigen expression and adaptive immune responses is an area of active research. Teasing apart these separate contributions is further complicated by the difference in innate immune sensing of LNP formulated mRNA in preclinical models compared to humans[46]. Nevertheless, Bergamaschi et al., demonstrated that a group of pro-inflammatory cytokines correlated with humoral responses following BNT162b2 vaccination[47]. Our findings also show a similar cluster of cytokines/chemokines of CXCL10, IL15, TNF, and CCL-3/4 in NHP, which are indicative of type I and II IFN response, upregulated by the SMARRT.RSV.preF. Although the pathway analysis confirmed this, the testing panel was specifically designed to identify analytes involved in the inflammatory process, which could possibly introduce bias in the data. Downregulated analytes included members of the Matrix Metalloprotease family such as MMP-12 and MMP-1 that are known to play an important role in regulating inflammatory response in macrophages[48]. Further investigation is required to identify possible immune cells targeted by SMARRT.RSV.preF at the site of immunization. A drawback with our study is the small sample size of our study which limited our ability to draw conclusions about the association between these analytes and adaptive immune responses. Nonetheless, these analytes were found to be associated with the immunogenicity of a saRNA vector in a preclinical model, albeit with a different LNP formulation[49]. Therefore, validating these markers in humans would be beneficial in understanding the connection between innate and adaptive immunity following saRNA vaccination.

In conclusion, our data support a strategy to use saRNA-based vaccines to boost pre-immune responses against RSV. Our results demonstrate this platform has the potential to enhance systemic and mucosal humoral responses in NHPs that were previously exposed to RSV. Furthermore, we show the polyfunctional nature of anamnestic memory T-cells generated by SMARRT.RSV.preF vaccine. The inherent ability of the saRNA platform to induce a Th1 response and boost immune responses with relatively low doses in both the naïve as well as pre-exposed setting individuals makes it an ideal platform for RSV vaccines in children and adults respectively.

## Methods
### Ethics statement
The mouse studies at Janssen Vaccines and Prevention B.V. were conducted in accordance with the Dutch Animal Experimentation Act and the Guidelines on the Protection of Animals for scientific purposes by

the Council of the European Committee. These studies were approved by the Centrale Commissie Dierproeven and the Dier Experimenten Commissie of Janssen Vaccines and Prevention B.V. Additionally, the NHP study was conducted at the Alpha Genesis test facility and was approved by the IACUC committee of Alpha Genesis (IACUC#22-11).

## Vaccines

A saRNA vaccine vector expressing RSV prefusion protein was generated by inserting the full-length, codon-optimized Fusion protein gene of RSV.A2 stabilized in the prefusion conformation downstream of the sub genomic promoter within the SMARRT cloning vector to create SMARRT.RSV.preF plasmid. The SMARRT cloning vector encodes the full-length, non-structural protein region of the TC83 attenuated strain of Venezuelan equine encephalitis virus (VEEV)[25]. Protein coding sequences are flanked at the 5' end by the T7 RNA polymerase recognition sequence, VEEV 5' untranslated region (UTR), the alphavirus downstream loop (DLP) motif and the ribosome skipping p2A consensus sequence; at the 3' end by the VEEV 3' UTR and encoded poly A segment. A Kozak sequence precedes the RSV F insert. SMARRT.RSV.preF saRNA was generated via in vitro transcription (IVT) using linearized SMARRT.2A plasmid as template, T7 RNA polymerase (Roche), and CleanCap AU (TriLink) as co-transcriptional capping reagent. Following IVT, template DNA was degraded by DNase treatment. RNA was purified via either silica column under denaturing conditions (mouse studies) or oligo dT affinity chromatography (NHP studies). Following purification, RNA was analyzed for integrity and homogeneity by capillary gel electrophoresis using a standard protocol (Agilent). The relative presence of RNA species was calculated using a distribution rate spreadsheet. Purified RNA was also analyzed for protein content by by NanoOrange® assay (Molecular Probes), and endotoxin content by turbidimetric Limulus Amebocyte Lysate (LAL) assay (Endosafe nexgen-PTS, Charles River Laboratories).

For in vivo assessments, saRNA was formulated in lipid nanoparticles. Briefly, lipids were dissolved in ethanol, and ionizable lipid ALC-0315, Cholesterol, DSPC, and DMG-PEG2000 were mixed. LNP formulations were prepared in a microfluidic device (NanoAssemblr Ignite, Precision NanoSystems) by combining this lipid mix with purified SMARRT.RSV.preF saRNA. Subsequently, LNP formulations underwent dilution and were dialyzed overnight at 4 °C against 20 mM Tris-HCl (pH 7.5) containing 10% sucrose. LNPs were further subjected to centrifugation to achieve the targeted concentration and then passed through a 0.22 μm filter. The final LNP formulations were stored at −80 °C and investigated for particle size (Zetasizer, Malvern), RNA encapsulation via Ribogreen assay (Invitrogen), and endotoxin levels using the Limulus Amebocyte Lysate (LAL) assay (Endosafe nexgen-MCS, Charles River Laboratories). The LNPs exhibited a size ranging between 70–90 nm, PDI < 0.2, EE > 95%, and endotoxin levels <5 EU/mg. After in vivo experiments, LNPs were retested and found to maintain the same physical properties.

A replication-incompetent, E1/E3-deleted recombinant adenoviral vector based on Ad26 was engineered using the AdVac® system to express the full-length codon-optimized F gene from the RSV-A2 strain, which was stabilized in its prefusion conformation through specific amino acid substitutions[50]. The RSV.preF protein (PRPM) was purified from a stable Chinese Hamster Ovary (CHO) cell line, which was created by transfecting DNA constructs into CHO cells grown in suspension via electroporation. Single-cell clones were then isolated using semi-solid media cloning, expanded, and ranked based on expression titer and product quality. For the selected lead clone, master and working cell banks were established[36]. Binding by a conformational dependent monoclonal antibody CR9501 was used to confirm expression of a properly folded RSV.preF antigen.

## Cell culture and SMARRT.RSV.preF transfection

The potency of LNP-formulated SMARRT.RSV.preF was assessed in BHK cells (ATCC CCL-10, Manassas, VA, USA following transfection and detection of RSV fusion protein expression. BHK cells were cultured using standard conditions. Expression of RSV fusion protein was confirmed by flow cytometry 24 h post transfection.

For flow cytometry analysis, cells were stained with Fixable Viability Dye 506 (Invitrogen, 65-0866-74) for 15 min at 4 °C, protected from light. Cells were then fixed using eBioscience Foxp3 Transcription Factor Staining Buffer Kit (Invitrogen, 00-5523-00) per manufacturer's instructions. Intracellular staining was performed with biotinylated anti-RSV antibody[51]. APC-Streptavidin (BioLegend, 405207) was used for secondary detection.

## Animals and Immunizations

Groups of 8 female BALB/c mice aged 6–8 weeks old were housed under biosafety level 2 (BSL-2), in-house animal facility, at 20–23 °C, 50–55% humidity with 12-hour light/dark cycles. The vaccines were administered intramuscularly in a total volume of 50mcL per hind leg (100mcL per animal). The study groups were immunized with 0.1, 1, or 10 mcg of SMARRT.RSV.preF on both day 0 and day 28. The control group received a single dose of $10^{10}$ VPs of Ad26.RSV.preF on day 0. Serum samples and spleens were collected from the mice for immunogenicity analysis.

In the study involving cynomolgus NHPs aged 6–7 years old ($n = 16$), of mixed gender, 12 animals were inoculated with RSV.A2 or RSV.Memphis37 strain at week −12. The animals were housed in pairs within an BSL-2 containment facility, maintained according to AAALAC and United States Department of Agriculture standards, with a 12-hour light/dark cycle. Each animal received $10^6$ PFU of virus in 1.1 mL of PBS, with 0.1 mL administered intranasally and 1 mL administered intratracheally[52]. After viral administration, the animals were kept in an upright position with their heads tilted back for at least 10 min. The animals were rested for 12 weeks before immunization, with intermittent bleeding to monitor RSV.F specific humoral responses. Prior to vaccination, the animals were divided into groups of 4. RSV primed animals were distributed based on week −8 RSV.postF antibody levels to ensure similar starting titers before vaccination (Supplementary Fig. 4). At week 0, the RSV-naïve group received 10 mcg of SMARRT.RSV.preF, while the other three groups, consisting of RSV pre-exposed animals, received 50 mcg of PRPM, 1 mcg, and 10 mcg of SMARRT.RSV.preF, respectively. The vaccines were administered intramuscularly (500mcL) into the deltoid muscle. Serum, whole blood, and nasal swabs were collected at different timepoints during the three-month follow-up period after the first dose. The groups that received SMARRT.RSV.preF as the primary dose were given a second dose (booster) at week 16 with 1 mcg of SMARRT.RSV.preF. The immunogenicity assessment was carried out until week 24.

## Virus neutralization assays

For both naïve and RSV-primed mice and cynomolgus NHP (CMs), virus neutralization assays were conducted in a similar manner. Heat-inactivated serum samples from the animals were diluted serially and mixed with 25,000 pfu of firefly luciferase (FFL)-labeled RSV CL57, which was cultured on A549 cells (ATCC CCL-185, Manassas, VA, USA). This mixture was incubated for 1 h at room temperature. Following that, 5000 A549 cells (multiplicity of infection: 5) were added to each well, and the plates were incubated for 20 h at 37 °C in 10% CO2. After the incubation period, Neolite substrate was added, and the luminescence signal was measured using an Envision® plate reader. The virus neutralization titers (VNT) were reported as the 50% inhibitory concentration (IC50).

## ELISA

The measurement of RSV preF-specific IgG antibodies in the serum of mice and cynomolgus NHP was performed using ELISA. 96-well half-area plates were coated with streptavidin (0.5 μg/mL) and left to incubate at 4 °C overnight. The wells were then washed with PBS containing 0.05% Tween-20 and blocked with 1% casein buffer in PBS for 1 h at room temperature (RT). After another round of washing, biotinylated RSV preF protein (2 μg/mL) was added to the wells and incubated for 1 h at RT. Following another wash, heat-inactivated serum samples and standards were added to the wells and incubated for 1 h at RT. RSV preF-specific antibodies were detected using horseradish peroxidase (HRP)-labeled anti-mouse IgG (1:5000, cat#1706516, Bio-Rad). The reaction was developed using LumiGLO® substrate, and the luminescence signal was measured at 428 nm. A four-parameter logistic (4PL) model was applied to the standard curve on each sample plate, and the titers were expressed as log10 relative potency compared to the standard serum sample.

In NHPs, the titers of RSV preF-specific IgA in nasal swabs were determined using ELISA. White half-area 96-well plates were coated with streptavidin (0.5 μg/mL) and incubated overnight at 4 °C. The wells were washed with PBS containing 0.05% Tween-20 and blocked with 1% casein buffer in PBS for 1 h at RT. After another wash, biotinylated RSV preF protein (2 μg/mL) was added and incubated for 1 h at RT. Following another wash, serially diluted heat-inactivated serum samples were added to the wells and incubated for 1 h at RT, followed by another wash. HRP-labeled anti-monkey IgA (1:5000, SAB3700759, Sigma-Aldrich) was added for detection, and the reaction was developed using LumiGLO® substrate. The luminescence signal was measured at 428 nm. A blank sample, included on every assay plate, was used to calculate endpoint titers. For nasal swabs, the total protein concentration was measured using OD280 measurement, which was used to correct for sample concentration. The RSV preF-specific IgA-binding titers in nasal swabs were expressed as log10 endpoint titers normalized to total protein concentration (mg/mL). Quality control serum samples were included in all runs.

## IFNγ ELISpot assay

For non-human primate studies (NHPs), EDTA-anticoagulated whole blood was shipped overnight at ambient temperature to Beth Israel Deaconess Medical Center (BIDMC). Peripheral blood mononuclear cells (PBMCs) were isolated using Ficoll-Paque and counted using a Guava EasyCyte Plus. The assay was performed in triplicates, with 200,000 cells per stimulation condition. Cells were stimulated with an overlapping peptide pool of the F protein from RSV.A2 (at a concentration of 2 mcg/mL) to detect antigen-specific responses, while medium stimulation was used to measure background response. As a positive control, cells were stimulated with phytohemagglutinin-M (PHA-M). A response was considered positive according to the assay positivity criteria if the background-subtracted values were greater than 50 SFU/10⁶ cells and exceeded two times the medium response for each animal. Any values below the threshold of assay positivity were set to 50 SFU in the figures and for statistical analysis.

## Intra-cellular cytokine staining assay

Cellular immune responses specific to the antigen were assessed using ICS on mouse splenocytes that were isolated on day 56. The splenocytes were stimulated with an overlapping peptide pool of the F protein from RSV.A2, hamster-anti-mouse CD28 (1:500, cat#553294, BD Biosciences) and rat-anti-mouse CD49d (1:500, cat#553153, BD Biosciences) for one hour, followed by the addition of BD GolgiPlug™. The samples were then incubated for 4 h at 37 °C and 5% CO₂ followed by overnight incubation at 4 °C and 5% CO₂. To identify dead cells, amine-reactive violet dye from Invitrogen was used for staining. Fc receptors were blocked using anti-mouse CD16/CD32 antibodies (1:50, cat#553142, BD Biosciences). The cells were further stained with anti-

CD3-FITC (1:400, Clone 142-2C11, cat#553062, BD Biosciences), CD4-PerCpCy5.5 (1:400, Clone RM4-5, cat#550954, BD Biosciences), and CD8-APC.H7 (1:75, Clone 53-6.7, cat#560182, BD Biosciences). BD Cytofix/Cytoperm™ was used for permeabilization, and intracellular staining was performed using IFNγ-PE (1:200, Clone XMG1.2, cat#554412, BD Biosciences), TNFα-PE.Cy7 (1:200, Clone MP6-XT22, cat#557644, BD Biosciences), and IL2-APC (1:300, Clone JES6-5H4, cat#554429, BD Biosciences) antibodies. Flow cytometry analysis of the percentage of live CD3⁺CD4⁺ and CD3⁺CD8⁺ T cells expressing IFN-γ, TNF-α, or IL-2 was conducted using a BD FACSCanto™ II flow cytometer. The gating strategy is illustrated in Supplementary Fig. 3. All the reagents used were from BD Biosciences, San Diego, CA, USA. FlowJo software version 9.6.1 (FlowJo, LLC, Ashland, OR, USA) was employed for the analysis of flow cytometric data.

For non-human primate ICS, cryopreserved PBMC were thawed and allowed to rest at 37 °C in a 5% CO₂ environment for 4 h. Afterward, the PBMC were incubated for 6 h at 37 °C in 5% CO₂ environment in the presence of: RPMI with 10% fetal bovine serum (unstimulated), Staphylococcus enterotoxin B (SEB) as a positive control, or peptide pool spanning the entire RSV.A2 fusion protein. All cultures had a protein transport inhibitor called monensin (GolgiStop; Becton, Dickinson and Company) and 1 μg/ml of anti-CD49d (Becton, Dickinson and Company, Cat# 340976). The cultured cells (1–2 million in 100 mcL) were then stained with a cell viability marker and predetermined quantities of antibodies (BD Pharmingen) against various markers such as CD3-R718 (2mcL; clone SP34.2; Cat#566955), CD4-BV711 (0.75 mcL; clone L200; Cat#563913), CD8-FITC (2.5 mcL; clone SK1; Cat#347313), CD45-BV786 (1.25 mcL; clone D058-1283; Cat#563861), CD95-PE.CF594 (0.25 mcL; clone DX2; Cat#562395), CD28-PE.Cy7 (1.25 mcL; clone 28.2; Cat#25-0289-42 from Life Technologies), CD107a-BV421 (2.5 mcL; clone H4A3; Cat#562623), TNFα-BV650 (3 mcL; clone Mab11; Cat#563418), IFNγ-BUV395 (2.5 mcL; clone B27; Cat#563563), and IL2-APC (0.3 mcL; clone MQ1-17H12; Cat#554567). Samples were only included if they had at least 1,000 viable CD4⁺ or CD8⁺ T cells. Finally, the samples were analyzed using an LSR II flow cytometer (Becton, Dickinson and Company, Franklin Lakes, NJ) with FlowJo v10.8.1 software.

The flow cytometry-generated polyfunctionality analysis and cell phenotype data sets were analyzed using SPICE 6.0 software (https://niaid.github.io/spice/), following the guidelines and considerations provided by the software developers[30]. Additionally, polyfunctionality scores (PFS) were determined with COMPASS (Combinatorial Polyfunctionality analysis of Antigen-Specific T-cell Subsets) according to the instructions provided by the authors[29]. PFS calculates a single number for each animal, summarizing the posterior probabilities of antigen-specific response across different cell subsets. The calculation takes into account the functionality levels of each subset, with a preference for subsets demonstrating higher degrees of functionality.

## Cytokine profiling

The Olink® Target 48 Cytokine panel was used to analyze pre-immunization and 24-hour post-immunization serum samples collected from NHPs. After expressing the analyte concentrations as pg/mL and log 10 transforming them, values outside the linear range were set at the upper and lower limits of quantification. Analytes in the treatment groups that did not have at least 50% of the samples above the lower limit of quantification were excluded from further analysis.

Principal component analysis, on scaled and centered data was performed using the 'prcomp' function in R. Statistical comparisons of analyte concentrations between treatment groups were conducted using a mixed effects model by the limma package. Specifically, mean differences per analyte between the 0 and 24-hour timepoints for the SMARRT.RSV.preF doses, irrespective of pre-exposure status, were estimated with group as fixed effect and analyzed, with animal ID as a

random effect and taking the estimated correlation between repeated measurements of paired samples from the same animal into account.

Spearman rank correlations were calculated between analyte measurements at 24 h for the SMARRT.RSV.preF 10 mcg dosage using the 'cor' function in R. The resulting correlation matrix was visualized using the corrplot package, with the ordering based on the first principal component.

Gene set enrichment analysis was performed using the 'camera' test from the limma package, with the default settings. The Molecular Signatures Database (MSigDB) hallmark gene set collection, imported via misgdbr, was used for the analysis. The entire analysis was conducted using R version 4.1.2[53], with the following package versions: limma 3.50.3[54], corrplot 0.92[55], and msigdbr 7.5.1[56].

### Statistical analysis

In the studies conducted on mice, statistical comparisons between different timepoints for each dose level of the vaccine were performed using analysis of variance (ANOVA) for potentially censored values (Tobit model). For the VNA and ELISA data, a log2 and log10 transformation was applied respectively.

In the non-human primate study, changes over time within groups (before and after immunization) were analyzed using ANOVA and a Bonferroni correction was applied to account for multiple comparisons. Similarly, exploratory comparisons were made between the group infected with RSV and the naïve group immunized with SMARRT.RSV.preF. For this analysis, ANOVA for potentially censored values (Tobit model) was used, and a Bonferroni correction was applied for multiple testing. The peak response for each individual readout was determined for each animal within a group to facilitate comparison. All statistical analyses were conducted using SAS version 9.4 (SAS Institute, Inc., Cary, NC, USA).

### Reporting summary

Further information on research design is available in the Nature Portfolio Reporting Summary linked to this article.

## Data availability

All of the final data has been included in main figures or supplementary information. Source data are provided with this paper.

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

## Author contributions

A.V., R.V., and R.Z. contributed to the conception of the work. A.V., R.G., S.K., and R.Z., designed studies. A.V., R.G., Y.J., M.V.K., S.J., S.B., M.B., H.V.D., H.K., J.R.V., A.B., L.N., S.R., B.J., J.A., M.L., G.M. B.M., and S.S. performed the experiments and analyzed the data. J.S. performed the statistical analysis. A.V. and R.Z wrote the original draft. A.V., R.V., R.G., S.K., Y.J., T.R.G., H.S., J.C., and R.Z edited the manuscript. All authors reviewed the manuscript.

## Competing interests

The authors declare the following competing interests: A.V., R.V., R.G., Y.J., S.K., M.V.K., S.J., S.B., M.B., H.V.D., H.K., J.R.V., A.B., L.N., S.R., B.J., J.A., J.S., T.R.G., H.S., J.C., and R.Z., are or were employees of Janssen Vaccines & Prevention B.V. or of Johnson & Johnson Innovative Medicine while engaged in the research project. These authors held or still hold stock in Johnson & Johnson. The remaining authors declare no competing interests.

## Additional information

Aneesh Vijayan ®[1,4] ✉, Ronald Vogels ®[1], Rachel Groppo[2], Yi Jin[2], Selina Khan ®[1,5], Mirjam Van Kampen ®[1], Sytze Jorritsma[1], Satish Boedhoe[1], Miranda Baert[1,6], Harry van Diepen[1], Harmjan Kuipers[1], Jan Serroyen[1], Jorge Reyes-del Valle[2], Ann Broman[2], Lannie Nguyen[2], Sayoni Ray[2], Bader Jarai[2], Jayant Arora[2], Michelle Lifton ®[3], Benjamin Mildenberg ®[3], Georgeanna Morton[3], Sampa Santra ®[3], Tamar R. Grossman[2], Hanneke Schuitemaker[1], Jerome Custers ®[1] & Roland Zahn ®[1] ✉

[1]Janssen Vaccines and Prevention B.V., Leiden, The Netherlands. [2]Johnson & Johnson Innovative Medicine, La Jolla, USA. [3]Center for Virology and Vaccine Research, Beth Israel Deaconess Medical Center, Harvard Medical School., Boston, USA. [4]Present address: Artemis Bioservices, Delft, The Netherlands. [5]Present address: Oncode Accelerator Foundation, Utrecht, The Netherlands. [6]Present address: LUCID research centre, Leiden Medical University, Leiden, The Netherlands. ✉e-mail: aneesh.vijayan@gmail.com; rzahn@its.jnj.com

