## [Peer review file · Nature Communications]

A self-amplifying RNA RSV prefusion-F vaccine elicits potent immunity in RSV pre-exposed and in naïve non-human primates.

Corresponding Author: Dr Aneesh Vijayan

Editorial Note: Parts of this peer review file have been redacted as indicated to avoid any copy right infringement.

Version 0:

Reviewer comments:

Reviewer #1

(Remarks to the Author)

The manuscript presents a self-amplifying RNA (SAM) vaccine based on pre-fusion F (pre-F). While three pre-F-based vaccines are already licensed, the SAM RNA vaccine in this study does not introduce significant modifications. The novelty of the vaccine composition is questionable, and the immunogenicity and effectiveness of the vaccine are not thoroughly verified. The control settings in several animal studies are unclear, and the rationale for developing a SAM RNA vaccine, given the existing vaccines, is not fully articulated. The authors should include comparisons of immunogenicity with marketed vaccines (at least with mRNA vaccines), evaluate vaccine immune persistence, challenge protection, and conduct VERD risk assessments.

Major Issues:

1. Figures 1B and 1C: The experiments need to be conducted at least three times. For Figure 1C, a non-functional SAM of the same length as the RNA vaccine should be used as a control to demonstrate that the observed effects are not attributable to dsRNA present in the SAM RNA itself. Furthermore, the dsRNA content after IVT should be assessed using ELISA. The authors described the transfection of BHK cells with the vaccine in line 130, but the method used was electrical transfer of SAM rather than LNP transfection of BHK cells. Please clarify the method used. If electroporation was employed, in vitro transfection results using LNP should also be included.
2. In the BALB/c mice and NHP immunogenicity studies, the authors used Ad26.RSV.preF or F protein without an adjuvant as controls. This is confusing given the availability of several RSV vaccines on the market. As an RNA vaccine, the authors should compare the immunogenicity with at least one mRNA vaccine. In NHP studies, clarify how the immunization dose for each vaccine group was determined.
3. In Figure 2D and Line 455, to conclude a Th1-biased immune response, the authors should test for the expression of at least one Th2 cytokine, such as IL-4.
4. Justify why the IgA test was conducted after the eighth week.
5. In the NHP experiments (Figure 3), all groups showed good immunogenicity with serum titers initially increasing and then decreasing. However, in Figure 4A, two monkeys in the 'PRPM (50mcg)' group showed almost no T cell response, and one monkey in the 'SMARRT.RSV.preF (1mcg)' group showed no detectable T cell response. Additionally, the 'PRPM (50mcg)' group showed a continuous decline in T cell response, inconsistent with the trend in Figure 3. Please explain these discrepancies.
6. Supplement the challenge results to demonstrate the protective efficacy of SMARRT.RSV.preF and assess the risk of vaccine-enhanced respiratory disease (VERD).

Minor Issues:

1. Line 20: Update to reflect the licensing of an RSV mRNA vaccine.
2. Line 60: Correct the duplication of "murine."
3. Line 65: Correct the repetition of "that that."
4. Line 83: Correct the redundancy in "based-based vaccine."
5. Line 143: Replace "Balb/C" with "BALB/c."
6. Figure 2B: Use NT90 instead of IC90.
7. Line 240: Clarify the missing data for the specified week.

8. Line 254: Remove the extra comma in "baseline,,"
9. Line 255: Correct the punctuation in "respectively.."
10. Line 285: Ensure there is a space before "with."
11. Line 330: Verify and correct the p-values; no $p < 0.0001$ groups are present.
12. Methods Section: Include details on the preparation and quality control of postF and the adenovirus vector vaccine. Display postF binding titers for each monkey clearly.
13. Line 554: Specify the antibodies used to control F protein expression.
14. Line 585: Correct to "10% CO₂."
15. Flow Cytometry: Supplement the gating strategy details.

Reviewer #2

(Remarks to the Author)

In this report, the authors test a self amplifying RNA vaccine for RSV (termed SMARRT.RSV.preF) in mice and cynomolgus macaques (both RSV naïve and previously RSV infected). Data collected with the self amplifying RNA vaccine are compared to a protein vaccine. They find a favorable immunogenicity profile with the SMARRT.RSV.preF vaccine, including induction of poly functional T cell responses and a Th1 skewed CD4 response, thought to lead to higher levels of neutralizing antibodies that may prevent development of enhanced disease as was found to be associated with an inactivated virus vaccine in the 1960s. Further, they find that vaccination of pre-immune NHP leads to development of robust anamnestic immune responses (in both blood and nose), suggesting this may also occur in pre-immune humans, relevant as many have been infected with RSV during their lifetime. The manuscript is clear and the experiments are focused and supportive of the stated conclusions. The downside to all of the presented data is, of course, the lack of an assessment of vaccine efficacy in a challenge model. Nonetheless, the data presented clearly suggest this vaccine candidate shows immense promise for use in combatting RSV disease in vulnerable populations. I have only one minor comment.

Do the authors have any data on viral or immune dynamics after RSV challenge in the animals used for the pre-immune studies? These data are not critical as the demonstration of anamnestic responses clearly makes the point that prior infection led to immune responses that are subsequently boosted. However, these data may be helpful for correlating magnitude (or functionality) of anamnestic responses.

Version 1:

Reviewer comments:

Reviewer #1

(Remarks to the Author)

Overall, the authors' responses in the rebuttal were unclear and did not adequately indicate the changes made. Many line numbers provided did not correspond to the revised manuscript, and the figure and panel numbers were inconsistent, making it difficult to follow the revisions. For a more efficient review, I suggest the authors clearly reference all changes with consistent line numbers, use track changes to highlight modifications, and, if possible, provide the previous version of the manuscript for comparison.

I agree that there is a need to develop new RSV vaccines, particularly given the observed rapid decline in the protection offered by existing vaccines. However, the manuscript does not provide sufficient evidence that the SAM vaccine offers any advantages over current vaccines in terms of long-term protection, either in immunogenicity or protective efficacy. The protective effects of the SAM vaccine have not been fully evaluated. Therefore, I maintain that it is essential to include at least one mRNA vaccine as a control to assess whether the SAM vaccine represents a promising new approach for RSV prevention.

Minor comment

12. Methods Section: Include details on the preparation and quality control of postF and the adenovirus vector vaccine. Display postF binding titers for each monkey clearly.

Response: We have added the requested information on the construction of Ad26.RSV.preF and RSV.preF protein in lines 547-554.

Response: In line 226, my request specifically refers to the preparation of the post-F protein, as the post-F titers are used for grouping the monkeys at this point, but the exact criteria for grouping are not provided.

Reviewer #2

(Remarks to the Author)

my minor comments have been addressed.

Reviewer #1 (Remarks to the Author):

The manuscript presents a self-amplifying RNA (SAM) vaccine based on pre-fusion F (pre-F). While three pre-F-based vaccines are already licensed, the SAM RNA vaccine in this study does not introduce significant modifications. The novelty of the vaccine composition is questionable, and the immunogenicity and effectiveness of the vaccine are not thoroughly verified. The control settings in several animal studies are unclear, and the rationale for developing a SAM RNA vaccine, given the existing vaccines, is not fully articulated. The authors should include comparisons of immunogenicity with marketed vaccines (at least with mRNA vaccines), evaluate vaccine immune persistence, challenge protection, and conduct VERD risk assessments.

We appreciate the reviewer's feedback. While there are existing licensed vaccines with good efficacy, long term follow up and revaccination studies after several years have not been performed at this point and only recently a study was published on the reduction of hospitalization in a vaccinated publication. In addition, the mRNA vaccine showed a more rapid drop in protection levels than the subunit-based vaccines, hence additional vaccine modalities are warranted to develop differentiated vaccines that may perform better. In this manuscript we explored the possibility of using a self-replicating RNA based vaccine for prevention of RSV and have performed a thorough immunological assessment. We have addressed several of the comments and provided explanations for the concerns raised.

Major Issues:

1. Figures 1B and 1C: The experiments need to be conducted at least three times. For Figure 1C, a non-functional SAM of the same length as the RNA vaccine should be used as a control to demonstrate that the observed effects are not attributable to dsRNA present in the SAM RNA itself. Furthermore, the dsRNA content after IVT should be assessed using ELISA. The authors described the transfection of BHK cells with the vaccine in line 130, but the method used was electrical transfer of SAM rather than LNP transfection of BHK cells. Please clarify the method used. If electroporation was employed, in vitro transfection results using LNP should also be included.

Response: Regarding Figure 1B, we have now incorporated data from two independent batches. This ensures that the results are consistent and reproducible across different production batches.

For saRNA vaccine, the assessment of dsRNA and its removal during purification might be less pronounced compared to conventional mRNAs. While purification can eliminate nonspecific dsRNAs, the generation of new dsRNA intermediates as part of the replication process is unavoidable. However, to avoid potential confusion for the readers we have removed Figure 1C.

Regarding the method of transfection described in line 130, we apologize for any confusion. We typically test unformulated saRNA using electroporation, while LNP-formulated saRNA is used for direct transfection of BHK cells. The data presented in Figure 1 reflects the transfection of BHK cells using formulated saRNA. This is clarified in the figure legend to avoid any misunderstanding (line 142).

2. In the BALB/c mice and NHP immunogenicity studies, the authors used Ad26.RSV.preF or F protein without an adjuvant as controls. This is confusing given the availability of several RSV vaccines on the market. As an RNA vaccine, the authors should compare the immunogenicity with at least one mRNA vaccine. In NHP studies, clarify how the immunization dose for each vaccine group was determined.

Response: We acknowledge the importance of including a licensed RSV vaccine in our study. However, at the time of the in-life phase of our study several vaccine candidates including Ad26.RSV.preF and pre.F protein were in development whereas no RSV vaccine was commercially available. Specifically, Moderna's RSV vaccine (mRESVIA) received FDA approval only on May 31, 2024, after the completion of our study (source: Moderna Receives U.S. FDA Approval for RSV Vaccine mRESVIA). Consequently, we utilized Ad26.RSV.preF and RSV.preF protein (PRPM) as controls in our study. These vaccines have been extensively tested in preclinical models, including non-human primates (NHPs), and importantly have shown efficacy in human clinical trials similar to the now licensed vaccines, as referenced in our manuscript (see references 26 and 38).

Regarding the immunization doses used in the NHP studies, we have clarified the rationale for the selected doses in the revised manuscript (lines 423). The doses were chosen based on previous preclinical and human studies with the same or different antigens, where no additional immunogenic benefit was observed with doses above 10 mcg in NHPs. This phenomenon is likely due to the stronger innate response triggered by the replicon if given at higher doses, which limits replication and thereby reduces antigen expression (see references 20 and 23).

3. In Figure 2D and Line 455, to conclude a Th1-biased immune response, the authors should test for the expression of at least one Th2 cytokine, such as IL-4.

Response: In our study, SMARRT immunization induced a type I IFN cytokine milieu, as evidenced by Olink analysis in addition to IL-15 (Fig 5C). Notably, IL-4 was undetectable in the serum of immunized animals, as levels remained below the lower limit of quantification (LLoQ) and therefore did not meet the criteria for inclusion in downstream analysis (Lines 696-698). This observation aligns with findings by Devarajan et al.; Cell Reports, 2023, which suggest that the development of cytotoxic CD4 T cells (derived from Th1 CD4 cells) requires both a type I IFN response and IL-15. This is consistent with our data showing that the proportion of memory CD4 cells contributing to IFN- γ and CD107a expression was higher in pre-exposed animals following SMARRT vaccination (Lines 302).

Taken together, our data indicate that the early cytokine milieu following SMARRT vaccination favored a Th1-skewed response. In addition, the IFN-g ELISpot assay clearly shows IFN-g induction only after SMARRT immunization in naïve and RSV pre-exposed animals, but not after protein immunization.

4. Justify why the IgA test was conducted after the eighth week.

Response: The decision to conduct the IgA test after the eighth week was primarily due to the sampling of nasal swabs. As noted in the *Materials and Methods* section, we excluded nasal swabs that were contaminated by blood, as this could interfere with the accurate assessment of

nasal IgA levels. Weeks 0 and 8 were the time points at which we had a sufficient number of uncontaminated samples available for analysis. We have clarified this in lines 244-245.

5. In the NHP experiments (Figure 3), all groups showed good immunogenicity with serum titers initially increasing and then decreasing. However, in Figure 4A, two monkeys in the 'PRPM (50mcg)' group showed almost no T cell response, and one monkey in the 'SMARRT.RSV.preF (1mcg)' group showed no detectable T cell response. Additionally, the 'PRPM (50mcg)' group showed a continuous decline in T cell response, inconsistent with the trend in Figure 3. Please explain these discrepancies.

Response: Figure 3 represents humoral immune responses (specifically RSV.preF binding and RSV neutralizing antibodies), while Figure 4 shows IFN- γ ELISpot data, reflecting cellular immune responses and these two types of immune responses can have different kinetic, dependent on vaccine platform used. For instance, there is evidence that adjuvanted RSV.preF vaccines induce significantly more CD4 T-cell responses compared to non-adjuvanted protein vaccines, while the humoral responses are not as strongly influenced by the adjuvant in older adults (Roels et al., JID, 2023).

We used outbred NHPs in our study, and similar to humans, cellular responses are more different in magnitude than humoral responses. Importantly, while the levels of IFN-g positive cells in all groups differ in the RSV infected NHP prior to vaccine dosage none of the PRPM dosed animals have an increase in the cellular response while 7 out of 8 SMARRT dosed animals show an increased response. Also, the naïve NHPs develop a cellular response after SMARRT dosing. The PRPM response is in line with an earlier naïve NHP study (reference 38), where we observed that Ad26.RSV.preF, which is a viral vector-based vaccine, elicited potent cellular responses. In contrast, PRPM, even at a higher dose of 150 mcg, did not elicit significant cellular responses (see Figure below from reference 38). This phenomenon might be due to the direct engagement of B cells by the soluble RSV.preF antigen (PRPM), as opposed to the membrane-bound antigen presentation following Ad26.RSV.preF or in current study by SMARRT vaccination.

[redacted]

We indeed observe a decline in the response level in the SMARRT.RSV.preF (10 mcg) group at week 12, however that is similar to the humoral immune response. A longer follow up time would have been needed to understand if the animals immune response would reach a certain setpoint after which a decline is minimal as observed with other vaccines.

6. Supplement the challenge results to demonstrate the protective efficacy of SMARRT.RSV.preF and assess the risk of vaccine-enhanced respiratory disease (VERD).

Response: Evaluating vaccine efficacy and safety in preclinical models (cotton rats) typically requires the use of naïve animals. However, this approach does not accurately reflect the immune dynamics of a pre-exposed population, which is our initially intended older adult target group. Additionally, unlike humans, RSV infection in preclinical animal models, such as NHPs, can induce long-term protection. This may mask the vaccine's efficacy when tested in a pre-exposed setting (Eyles et al.; JID, 2013). Furthermore, in the gold standard RSV model the cotton rat, Geall et al.; PNAS, 2012 observed for both LNP-formulated and naked saRNA encoding a wild type fusion protein of RSV protection at a 1 mcg dose. This suggest that the cotton rat model might not be sensitive enough to accurately evaluate the efficacy of highly potent saRNA based RSV vaccines. In addition, only the advent of prefusion stabilized RSV F protein led to the success of multiple vaccine in human efficacy trials clearly showing that functional pre-F specific antibodies are required for protection from RSV infection in humans. Showing induction of such antibodies in animals are therefore considered to be a good surrogate for protection, reducing the need for confirmatory animal efficacy studies, instead of further clinical development.

VERD is mainly a recognized risk in young RSV seronegative infants and primarily attributed to the induction of poorly neutralizing antibodies by RSV vaccines, especially inactivated virus, which also leads to a Th2-skewed immune response. This is a significant risk for direct RSV naïve infant vaccination with poorly neutralizing and Th2 bias immunity inducing vaccines. However, recent studies, such as the one by (Eichinger et al.; Front. Immunol., 2022), have shown that prior exposure to RSV in neonatal mice mitigated vaccine-induced Th2-skewed CD4 T-cell responses and the associated IL-13⁺ and IL-5⁺ ILC2 responses, which are linked to mucus production and lung inflammation. Given that our target population are pre-exposed individuals, the risk of VERD is minimal and has not been described to date for RSV despite larger efficacy trials with vaccines that did not induce neutralizing antibodies to a significant extend.

Overall, the relevance of testing efficacy and VERD in a pre-exposed preclinical setting is limited, and the results are unlikely to be directly extrapolatable to humans. Therefore, we believe that conducting such studies would not provide meaningful insights into the vaccine's performance in the intended population.

Minor Issues:

1. Line 20: Update to reflect the licensing of an RSV mRNA vaccine.

Response: Added. See line 20.

2. Line 60: Correct the duplication of "murine."

Response: Done. See line 65.

4. Line 83: Correct the redundancy in "based-based vaccine."

Response: Done. See line 83

5. Line 143: Replace "Balb/C" with "BALB/c."

Response: Done. See lines 145, 168 and 574.

6. Figure 2B: Use NT90 instead of IC90.

Response: In our study, we utilized the 50% inhibitory concentration (IC50) as the standard unit in our established assay, consistent with our previous published work (ref 38, Freek et al., Vaccines, 2023). Using IC50 allows for easier comparison across studies, ensuring consistency and facilitating the interpretation of our results in the context of existing research.

7. Line 240: Clarify the missing data for the specified week.

Response: Added. See lines 244-245.

8. Line 254: Remove the extra comma in "baseline,,"

Response: Done. See line 259.

9. Line 255: Correct the punctuation in "respectively.."

Response: Done. See line 260.

10. Line 285: Ensure there is a space before "with."

Response: Done. See line 291.

11. Line 330: Verify and correct the p-values; no $p < 0.0001$ groups are present.

Response: Corrected

12. Methods Section: Include details on the preparation and quality control of postF and the adenovirus vector vaccine. Display postF binding titers for each monkey clearly.

Response: We have added the requested information on the construction of Ad26.RSV.preF and RSV.preF protein in lines 547-554.

13. Line 554: Specify the antibodies used to control F protein expression.

Response: We used a preF conformational specific antibody (line 555).

14. Line 585: Correct to "10% CO2."

Response: Corrected.

15. Flow Cytometry: Supplement the gating strategy details.

Response: The gating strategy is illustrated as a supplementary figure (Fig S3; line 664)

Reviewer #2 (Remarks to the Author):

In this report, the authors test a self amplifying RNA vaccine for RSV (termed SMARRT.RSV.preF) in mice and cynomolgus macaques (both RSV naïve and previously RSV infected). Data collected with the self amplifying RNA vaccine are compared to a protein vaccine. They find a favorable immunogenicity profile with the SMARRT.RSV.preF vaccine, including induction of poly functional T cell responses and a Th1 skewed CD4 response, thought to lead to higher levels of neutralizing antibodies that may prevent development of enhanced disease as was found to be associated with an inactivated virus vaccine in the 1960s. Further, they find that vaccination of pre-immune NHP leads to development of robust anamnestic immune responses (in both blood and nose), suggesting this may also occur in pre-immune humans, relevant as many have been infected with RSV during their lifetime. The manuscript is clear and the experiments are focused and supportive of the stated conclusions. The downside to all of the presented data is, of course, the lack of an assessment of vaccine efficacy in a challenge model. Nonetheless, the data presented clearly suggest this vaccine candidate shows immense promise for use in combatting RSV disease in vulnerable populations. I have only one minor comment.

We appreciate the reviewer's assessment, noting that "the data presented clearly suggest this vaccine candidate shows immense promise for combating RSV disease in vulnerable populations." The minor comment raised by the reviewer has been addressed below.

Evaluating vaccine efficacy and safety in preclinical models (cotton rats) typically requires the use of naïve animals. However, this approach does not accurately reflect the immune dynamics of a pre-exposed population, which is our initially intended older adult target group. Additionally, unlike humans, RSV infection in preclinical animal models, such as NHPs, can induce long-term protection. This may mask the vaccine's efficacy when tested in a pre-exposed setting (Eyles et al.; JID, 2013). Furthermore, in the gold standard RSV model the cotton rat, Geall et al.; PNAS, 2012 observed for both LNP-formulated and naked saRNA encoding a wild type fusion protein of RSV protection at a 1 mcg dose. This suggest that the cotton rat model might not be sensitive enough to accurately evaluate the efficacy of highly potent saRNA based RSV vaccines. In addition, only the advent of prefusion stabilized RSV F protein led to the success of multiple vaccine in human efficacy trials clearly showing that functional pre-F specific antibodies are required for protection from RSV infection in humans. Showing induction of such antibodies in animals are therefore considered to be a good surrogate for protection, reducing the need for confirmatory animal efficacy studies, instead of further clinical development.

Do the authors have any data on viral or immune dynamics after RSV challenge in the animals used for the pre-immune studies? These data are not critical as the demonstration of anamnestic responses clearly makes the point that prior infection led to immune responses that are subsequently boosted. However, these data may be helpful for correlating magnitude (or functionality) of anamnestic responses.

Response: We have provided additional data in the supplementary information that addresses the immune dynamics following RSV challenge. Specifically, we measured binding antibodies

against the post-fusion protein of RSV (RSV.postF) in the challenged animals. As shown in Figure S4, all animals infected with RSV became seropositive within two weeks post-challenge.

Reviewer #1 (Remarks to the Author):

Overall, the authors' responses in the rebuttal were unclear and did not adequately indicate the changes made. Many line numbers provided did not correspond to the revised manuscript, and the figure and panel numbers were inconsistent, making it difficult to follow the revisions. For a more efficient review, I suggest the authors clearly reference all changes with consistent line numbers, use track changes to highlight modifications, and, if possible, provide the previous version of the manuscript for comparison.

For clarity, we provide the line numbers in the latest version of the manuscript. To simplify tracking our revisions, we have also attached a PDF document.

We appreciate your patience, and we hope these adjustments make the review process more efficient. Thank you again for your constructive feedback.

P.S. For clarity, please refer to the corrected line numbers provided alongside each comment from 1st review round at the end of this rebuttal letter.

I agree that there is a need to develop new RSV vaccines, particularly given the observed rapid decline in the protection offered by existing vaccines. However, the manuscript does not provide sufficient evidence that the SAM vaccine offers any advantages over current vaccines in terms of long-term protection, either in immunogenicity or protective efficacy. The protective effects of the SAM vaccine have not been fully evaluated. Therefore, I maintain that it is essential to include at least one mRNA vaccine as a control to assess whether the SAM vaccine represents a promising new approach for RSV prevention.

Response: We addressed this concern in our previous rebuttal, noting the challenges associated with including an mRNA vaccine as a control. Additionally, while certain vaccines may have regulatory approval, this does not always translate to immediate or broad accessibility. As illustrated by our own experience trying to procure mRNA-based COVID-19 vaccines in 2023—despite their approval in 2021—availability can remain a significant hurdle. Thank you again for your valuable feedback.

Minor comment

12. Methods Section: Include details on the preparation and quality control of postF and the adenovirus vector vaccine. Display postF binding titers for each monkey clearly.

Response: We have added the requested information on the construction of Ad26.RSV.preF and RSV.preF protein in lines 547-554.

Response: In line 226, my request specifically refers to the preparation of the post-F protein, as the post-F titers are used for grouping the monkeys at this point, but the exact criteria for grouping are not provided.

Response: The specific criteria for grouping have been detailed in line 491 of the manuscript.

Reviewer #2 (Remarks to the Author):

my minor comments have been addressed.

Reviewer #1 (from 1st review):

Major Issues:

1. Figures 1B and 1C: The experiments need to be conducted at least three times. For Figure 1C, a non-functional SAM of the same length as the RNA vaccine should be used as a control to demonstrate that the observed effects are not attributable to dsRNA present in the SAM RNA itself. Furthermore, the dsRNA content after IVT should be assessed using ELISA. The authors described the transfection of BHK cells with the vaccine in line 130, but the method used was electrical transfer of SAM rather than LNP transfection of BHK cells. Please clarify the method used. If electroporation was employed, *in vitro* transfection results using LNP should also be included.

Response: (line 858).

2. In the BALB/c mice and NHP immunogenicity studies, the authors used Ad26.RSV.preF or F protein without an adjuvant as controls. This is confusing given the availability of several RSV vaccines on the market. As an RNA vaccine, the authors should compare the immunogenicity with at least one mRNA vaccine. In NHP studies, clarify how the immunization dose for each vaccine group was determined.

Response: (lines 330).

3. In Figure 2D and Line 455, to conclude a Th1-biased immune response, the authors should test for the expression of at least one Th2 cytokine, such as IL-4.

Response: (lines 604).

4. Justify why the IgA test was conducted after the eighth week.

Response: (lines 894).